# Digital Pathology for Better Clinical Practice

**DOI:** 10.3390/cancers16091686

**Published:** 2024-04-26

**Authors:** Assia Hijazi, Carlo Bifulco, Pamela Baldin, Jérôme Galon

**Affiliations:** 1The French National Institute of Health & Medical Research (INSERM), Laboratory of Integrative Cancer Immunology, F-75006 Paris, France; assia.hijazi@sorbonne-universite.fr; 2Equipe Labellisée Ligue Contre le Cancer, F-75006 Paris, France; 3Centre de Recherche des Cordeliers, Sorbonne Université, Université Paris Cité, F-75006 Paris, France; 4Providence Genomics, Portland, OR 02912, USA; carlo.bifulco@providence.org; 5Earle A Chiles Research Institute, Portland, OR 97213, USA; 6Department of Pathology, Cliniques Universitaires Saint Luc, UCLouvain, 1200 Brussels, Belgium; pamela.baldin@uclouvain.be; 7Veracyte, 13009 Marseille, France

**Keywords:** digital pathology (DP), immunoscore (IS), whole-slide imaging (WSI), artificial intelligence (AI), quantitative analysis, histopathology, colorectal cancer (CRC), diagnosis, biomarkers, clinical practice

## Abstract

**Simple Summary:**

This review highlights the profound impact of digital pathology (DP) and artificial intelligence (AI) on advancing cancer diagnosis and treatment. DP enables pathologists to access, analyze, and share high-resolution images, enhancing diagnostic accuracy and fostering remote collaboration. AI further refines cancer diagnosis by automating tasks and facilitating spatial analysis of the tumor microenvironment (TME), leading to the discovery of novel biomarkers. Immunoscore (IS), an AI-assisted immune assay, exhibits robust potential in improving cancer diagnosis, prognosis, and treatment selection, surpassing traditional staging systems. Integrating DP and AI, particularly the IS biomarker, into clinical practice promises to enhance personalized cancer therapy. The research underscores a pivotal leap forward in pathology, stressing the imperative of incorporating AI-driven technologies for improved cancer patient care and outcomes. This exploration aims to provide insights into the transformative potential of DP in cancer management, influencing the clinical community towards more effective diagnostic and therapeutic strategies.

**Abstract:**

(1) Background: Digital pathology (DP) is transforming the landscape of clinical practice, offering a revolutionary approach to traditional pathology analysis and diagnosis. (2) Methods: This innovative technology involves the digitization of traditional glass slides which enables pathologists to access, analyze, and share high-resolution whole-slide images (WSI) of tissue specimens in a digital format. By integrating cutting-edge imaging technology with advanced software, DP promises to enhance clinical practice in numerous ways. DP not only improves quality assurance and standardization but also allows remote collaboration among experts for a more accurate diagnosis. Artificial intelligence (AI) in pathology significantly improves cancer diagnosis, classification, and prognosis by automating various tasks. It also enhances the spatial analysis of tumor microenvironment (TME) and enables the discovery of new biomarkers, advancing their translation for therapeutic applications. (3) Results: The AI-driven immune assays, Immunoscore (IS) and Immunoscore-Immune Checkpoint (IS-IC), have emerged as powerful tools for improving cancer diagnosis, prognosis, and treatment selection by assessing the tumor immune contexture in cancer patients. Digital IS quantitative assessment performed on hematoxylin–eosin (H&E) and CD3+/CD8+ stained slides from colon cancer patients has proven to be more reproducible, concordant, and reliable than expert pathologists’ evaluation of immune response. Outperforming traditional staging systems, IS demonstrated robust potential to enhance treatment efficiency in clinical practice, ultimately advancing cancer patient care. Certainly, addressing the challenges DP has encountered is essential to ensure its successful integration into clinical guidelines and its implementation into clinical use. (4) Conclusion: The ongoing progress in DP holds the potential to revolutionize pathology practices, emphasizing the need to incorporate powerful AI technologies, including IS, into clinical settings to enhance personalized cancer therapy.

## 1. Introduction

### 1.1. The Forthcoming Transition of Traditional Pathology into the Digital Era

Digital pathology (DP), often perceived as a modern innovation, has its roots dating back several decades. The foundations of DP can be traced to the 1960s when Prewitt and Mendelsohn pioneered the scanning of microscopic images from blood smears [1]. This scanning method not only facilitated the identification of various cell types but also retained the spatial information of the analyzed blood samples. Over the past two decades, DP images have gained extensive use in medicine. A transformative advancement in the field was in the 1990s with the introduction of whole-slide imaging (WSI). This approach revolutionized the potential to scan entire tissue sections on slides, departing from the conventional practice of singling out specific regions of interest (ROI) for analysis. This progress was made possible with high-resolution WSI devices, capable of quickly digitizing classical glass slides at resolutions as fine as 0.23–0.25 μm per pixel. Pathologists from different locations can then review these digital images on a computer monitor [2,3] and collaborate on specific cases for second opinions.

Termed “digital pathology”, this innovation streamlines the extraction, management, and interpretation of patient histopathological data. Its primary objective is to tackle the common challenges faced by traditional pathologists by providing the capability to access and share scanned slide images, which facilitates remote clinical case diagnosis [4,5].

In traditional anatomical pathology, histopathologists manually assess and classify patients or diseases under a microscope. Thus, in clinical routine practice, pathologists rely on the visual evaluation and semi-quantification of morphological features in analyzed samples. While cost-effective, widely available, and compatible with formalin-fixed, paraffin-embedded (FFPE) tissue samples [6], these methods are susceptible to subjective interpretations, particularly in the evaluation of immune cells. This can potentially lead to inter-observer discrepancies among different pathologists and medical centers [7], affecting result reproducibility and consequently patients’ treatment decisions [8,9,10]. Remarkably, digital WSIs have demonstrated strong diagnostic agreement, often outperforming pathologists’ traditional light microscopy slide analysis and greatly facilitating the sharing of diagnostic slides among pathologists [11,12,13,14,15]. Furthermore, by enabling the wide application of artificial intelligence (AI), DP is contributing to breakthroughs in diagnostic potential and workflow efficiency. AI, a concept that emerged in the mid-20th century, aims to replicate human cognitive abilities in machines [16]. Within the realm of AI, machine learning (ML) methods focus on training machines from available data to enable predictive capabilities, like forecasting therapy responses or predicting cancer recurrence risks. Deep learning (DL), is a subset of ML emerged in the 1980s, and utilizes multi-layer neural networks to process data in a way that mimics human neural connections [17]. Essentially, these AI approaches are designed to extract meaningful image representations, which are subsequently processed by specific machine classifiers, based on specific criteria (segmentation, diagnostics, or prognostics) using supervised or unsupervised methods [18,19].

In fact, automated AI-based DP applied to hematoxylin–eosin (H&E)-stained slides can integrate multiple quantitative datasets to capture the tissue complexity and spatial cell organization of the TME, enhancing the automatic selection and analysis of ROIs while improving accuracy, reproducibility, and efficiency. Consequently, this has the potential to reduce human errors by lessening the reliance on subjective visual assessments by pathologists and may pave the way into a digitally empowered era in pathology and diagnosis of diseases, ultimately improving patient care.

This review provides an insight into the implementation of AI in DP and its potential to enhance current clinical practice. It will delve into the significance of Immunoscore (IS) and Immunoscore Immune-Checkpoints (IS-IC) as innovative AI-assisted DP assays, not only for diagnosing cancer patients and predicting their clinical outcomes but also for stratifying them based on response to treatment. Additionally, it will shed light on the substantial challenges that currently impede the adoption of AI-based tools in clinical settings.

### 1.2. Digital Pathology Empowers Quantitative Analysis of Whole-Slide Images (WSI)

In DP, WSIs can be analyzed with remarkable precision, significantly expediting the quantification of specific cell populations within tumors or distinct tissue types. Additionally, DP enables the identification and assessment of precise histological patterns and morphological features. Furthermore, DP exhibits the potential to pinpoint and precisely define ROI, such as distinguishing between tumoral and stromal areas or assessing the tumor core versus the invasive margin. Moreover, it can deeply explore and analyze cell-to-cell distances and interactions, leading to a better understanding of the tumor tissue microenvironment [19,20].

Quantitative image analysis provides a means to extract highly specific data from WSI, an incredibly labor-intensive and time-consuming task when executed manually. The integration of AI in a DP workflow holds the potential to address several challenging issues frequently encountered by pathologists, including tasks like visual cell counting across entire tissue slides and the assessment of multiple tumor parameters, which are prone to significant human errors [21,22]. Over the past few years, open-source WSI analysis software, coupled with robust clinical applications, has been developed [23,24]. These software solutions encompass tools for annotation and cell detection, and their affordability has led to a broader adoption of DP in numerous hospitals worldwide. However, the widespread implementation of DP and the significant investments in its clinical applications are still limited in many hospitals and clinical centers (https://www.rcpath.org/profession/digital-pathology.html (accessed on 1 February 2024)). This can be attributed to factors such as high initial costs of infrastructure setup, concerns regarding data security and privacy, challenges in staff training and workflow integration, and regulatory hurdles related to validation and accreditation. However, high-quality WSI databases are expected to emerge with the potential increase in DP centers [25]. This will enhance the development of algorithms using computer software to analyze WSIs for the study of diseases.

## 2. Clinical Applications of AI and DP

### 2.1. AI-Based Digital Pathology, a Powerful Driving Force in Cancer Research and Therapy

AI applications in pathology employ advanced image processing to extract precise information, recognize specific tissue patterns and identify biomarkers’ expression [26,27]. In cancer, AI-based approaches play a vital role in patient diagnosis, classification, and prognosis for treatment response. For instance, the algorithms designed for early tumor detection are instrumental in improving patient survival [28] and the potential of DP in cancer diagnosis is highly encouraging [29,30,31,32,33,34,35,36,37]. Additionally, DP addresses various challenges in oncology, including the development of prognostic assays to assess disease severity and predict tumor recurrence [38] and clinical outcomes in cancer patients [39,40].

DL algorithms are also being employed for the evaluation of Immunohistochemistry (IHC) and H&E-stained slides across various tumor types. Currently, their utilization in cancer research is extensive and includes identifying biomarkers that forecast early melanoma survival [41], recognizing invasive regions in breast cancer tissue [42], and predicting patient responses to chemotherapy in advanced rectal cancer [43].

In addition to T-helper 1 immune signature [44,45,46], germline genetics [47] or microbiomes [48] contributing to the immune landscape, there is growing evidence about the spatial organization of immune cells in the tumor microenvironment (TME) and its profound influence on cancer development [44]. DP revealed crucial insights into the spatial organization of immune cells within the TME, demonstrating its clinical relevance in various cancers [45,49]. In colorectal tumors, increased CD3+ T-cell density in the invasive margin strongly correlates with disease-free survival (DFS) [45], providing valuable information on immune cell interactions and their spatial relationships. DP also enhances our understanding of the dynamics of TME, immune cell phenotypes, and treatment resistance mechanisms [50] at single-cell resolution. This strategy offers valuable insights into intra-tumor heterogeneity and its predictive outcomes [51] when combined with spatial multi-omics data.

Additionally, tracking the spatio-temporal dynamics of immune cells during tumor progression is vital for gaining deeper insights into cancer’s immune control mechanisms. Thereby, the potential for spatial analysis of the TME has been greatly advanced through the application of DP to multiplex IHC imaging. The spatial quantitative analysis of high-plex images reveals co-localization of markers and immune cell infiltration across different tumor regions, offering insights into the TME’s architecture [52]. This multispectral approach also provides a deeper understanding of cell-specific gene expression [53], and the cellular relationships within tumor tissue.

Multiplexing also enabled the identification of T-cell populations and their relative spatial distributions in the TME. It also emphasized the potential of immune parameters, such as IS-IC to stratify patients who may benefit from immunotherapy [30,32,54,55]. In addition to IS, analysis of other immune cell subpopulations such as Th1 cells, Tfh cells, B-cells and macrophages yields valuable supplementary insights [44,46,51,56]. Whole-slide digital assessment, coupled with multiplexed imaging, presents an excellent opportunity to delve into cell signaling pathways in the TME by examining relative spatial cell distances. Notably, even when applied to specific regions of interest, this type of analysis can be challenging for pathologists to perform manually [57,58].

IHC is instrumental in characterizing biomarkers for therapy response [59]. However, enhancing precision in quantifying immune biomarkers, especially spatial cell phenotypes, requires advanced quantitative techniques. Also, the development of robust statistical methods for mapping TME components is crucial to expedite the understanding of intra-tumor heterogeneity and facilitate the translation of cancer biomarkers for therapeutic applications.

### 2.2. Translating Digital Pathology into Clinical Practice: Immunoscore and Immunoscore-IC, Novel Paradigms for Cancer Treatment

Tumor growth involves complex spatio-temporally regulated interactions between different TME components [60]. Recent advances in TME spatial analysis, driven by AI technologies, have identified new prognostic factors by assessing existing immune cell types [45]. This contributed to the prediction of responses to immune checkpoint inhibitors (ICIs) immunotherapy, notably highlighting the predictive value of tumor-infiltrating lymphocytes (TILs) in various cancers [61]. These insights bridge the gap to the emerging field of tumor profiling and precision oncology, emphasizing the need for predictive assays to stratify patients for personalized treatments. The analytical prowess of AI holds the potential to accelerate the discovery of novel histopathological biomarkers, predicting disease progression, tumor recurrence, and treatment response [18,25]. Recent years have seen increased AI implementation in histological imaging analysis, revolutionizing personalized therapy steps, including tumor detection, staging, and molecular subtyping for enhanced patient care. For instance, AI facilitates early personalized treatment in CRC by detecting specific biomarkers like MSI [62] that guide clinicians in selecting patients for specific therapies, including ICIs and targeted therapies. AI-enabled personalized cancer diagnosis and treatment are reshaping oncology with impactful examples. The integration of AI into Glioblastoma imaging holds promise for advancing the characterization and tracking of the disease, including recurrence. This integration has the potential to substantially enhance patient outcomes through enabling more accurate diagnosis, precise treatment planning, and improved monitoring of treatment response [63]. In ovarian cancer, AI algorithms predict chemotherapy responses based on tumor genomic profiles, guiding clinicians in selecting the most effective treatment regimen [64]. Furthermore, in pancreatic cancer, AI is transforming pancreatic cancer care, enhancing diagnostics, personalizing treatments, and optimizing operational efficiency, ultimately leading to improved patient outcomes [65]. Additionally, AI-powered liquid biopsy platforms in CRC holds promise for screening, stratifying patients for treatment, and real-time monitoring of treatment response, offering improved management and personalized care [66]. These examples highlight how AI empowers clinicians with actionable insights, leading to more personalized and effective cancer care strategies. Moreover, these DP practices can ultimately help patients avoid unnecessary treatment expenses and undesirable side effects.

DP can also evaluate composite biomarkers, including CD8 and PDL-1 in patients treated with anti-PDL-1 ICI [32,67]. These biomarkers have significant potential in predicting patients’ outcome and response to ICIs. Indeed, patients who are positive for CD8 and PDL-1 have shown prolonged survival compared with CD8 and PDL-1 negative patients.

Research on human cancers underscored the profound influence of the TME on cancer development. DP-assisted analysis of immune landscapes provided clinical insights, revealing the significance of immune cell density and location for patient survival. Automated image analysis by DP highlighted the crucial role of immune parameters in cancer therapy. This paved the way for innovative approaches like IS and IS-IC that can potentially revolutionize cancer diagnosis, prognosis, and treatment with objective and standardized AI-assisted assessments of the immune TME.

IS is a DP-immune assay originally designed to quantify CD8+ and CD3+ T lymphocytes within the TME to define T-cell abundance and infiltration in the tumor (Figure 1). IS provides a powerful prognostic tool as it shows high accuracy in defining tumor immune contexture in cancer patients by examining the density and location of immune cells [45]. IS also revealed that intratumoral adaptive immune reaction has a significant impact on patients’ survival [68,69] and demonstrated a robust prognostic value disregarding the tumor stage, by stratifying patients according to their tumor immune components.

This DP tool features high importance in sorting the patients at risk of tumor recurrence [33,34,35,70,71,72] (Figure 2A,B), and also in predicting their outcomes and response to treatments [73]. In colorectal cancer (CRC), IS assessment of infiltrated immune cells on FFPE-resected tumor sections was a powerful predictor of tumor recurrence (Figure 2B). IS detected a low adaptive immune reaction in the primary tumor of recurrent patients. The predictive values of IS have been validated on large cohorts of early and late stages CRC [33,34,35,36,74] and in randomized phase 3 clinical trials [75,76,77,78]. The prognostic value of IS has been also assessed in response to chemotherapy in all CRC stages [33,34,35,79] in response to neoadjuvant chemotherapy in bladder [80] and breast [81] cancer and to radiochemotherapy in rectal cancer [82,83,84]. Thus, these immune features, associated with cancer progression and risk of recurrence, revealed high clinical significance, especially guiding treatment-decisions in cancer patients.

Recently, the prediction of response to immunotherapy based on the immune contexture parameters, including IS, has been established [30,37]. For this aim, a novel DP-based assay, the IS-IC, has been developed to predict response to ICI immunotherapies. IS-IC is revealed as powerful predictive biomarker for response to combination immunotherapy in metastatic CRC patients, enrolled in phase II AtezoTRIBE clinical trial [30] and for response to anti-PDL-1 immunotherapy in non-small cell lung cancer (NSCLC) [32,55]. The latest findings also demonstrated that tumor immune contexture is a determinant of anti-CD19 CAR-T cell efficacy in large B-cell lymphoma (LBCL). It has also been proved that IS was robustly associated with response to CAR T-cell, as well as with prolonged patient survival [37]. This study driven by IS advanced the understanding of TME features associated with clinical responses to CAR-T cell therapy, which could optimize LBCL patient treatment.

The prognostic value of IS has now been assessed in a broad range of tumor types [29,37,80,85,86,87,88]. Moreover, the consensus IS has been validated worldwide for its powerful prognostic significance that outperforms the classical TNM tumor progression and invasion staging system [36,45,74]. Therefore, IS has shown that immune parameters of cancer patients are more powerful than the TNM classification system in predicting outcomes. This evidence would have not been possible through a visual assessment of the immune response by traditional pathology. Thus, IS and IS-IC can profoundly reshape the landscape of cancer diagnosis, prognosis, and treatment. Their potential lies in advancing the field of oncology, empowering patients with more precise and highly effective therapeutic options.

### 2.3. Immunoscore: A Reliable and Consistent Assay Surpassing Pathologists’ Visual Assessment

For many decades, conventional pathology techniques have played a crucial role in diagnosing and classifying cancer patients. However, the emergence of ground-breaking advances in precision medicine marked a paradigm shift in the development of DP-based methods for precise quantitative analyses.

IS has emerged as a powerful tool and a cutting-edge approach in the field of IHC and pathology, offering numerous advantages over pathologists’ visual evaluation and quantitative analyses of the immune response [89,90]. This digital assay ensures highly reproducible and consistent results, surpassing human visual assessments [89,90] (Figure 3A,B).

Comparisons between IS and pathologists’ visual scoring revealed significant discrepancies. A recent multi-institutional study highlighted the reproducibility and concordance of IS compared to non-concordant evaluations by pathologists in over 92% of cases. IS demonstrates high reproducibility, while agreement among pathologists remains weak, even after training [90], leading to notable misclassifications by pathologists, potentially impacting a substantial number of cases [90]. This non-concordance may misclassify up to 70% of cases, potentially affecting 87,000 colon cancer cases annually, which would potentially receive inadequate treatments [90]. IS, driven by DP, provides reliable diagnosis and patient stratification, revealing the prognostic significance of the immune response in treating CRC patients [34,35,79,89,90]. Thus, an assessment of stage II and III CC patients that does not included a DP IS quantification can drive suboptimal treatment decision-making [33,34,75,77,78,90]. High-risk patients may also benefit from different oncology treatments [91,92,93,94,95,96,97,98,99,100,101,102,103,104,105,106,107,108]. Therefore, standardized IS outperforms expert pathologists’ visual assessments, emphasizing its potential in improving cancer diagnosis and enabling personalized therapeutic decisions for CRC patient care.

### 2.4. Adoption of Digital Pathology and AI in Clinical Practice: Challenges, Limitations and Future Perspectives

The implementation of advanced DP and AI technologies in clinical practice is essential due to a shortage of pathologists and increased demographics driving (aging population) the cancer clinical diagnostic load. This approach improves efficiency, quality, and enhances overall diagnostic accuracy and collaboration among pathologists, clinicians, and researchers. Automated DP offers advantages such as a reduction in inaccuracies, faster analysis, and generation of high-throughput data, often beyond human visual analysis capabilities. This can significantly alleviate pathologists’ workload and enhances diagnostic accuracy [109].

However, despite the promising results of AI-powered digital pathology (AI-DP) and its potential in predicting histopathological diagnoses, the journey from algorithm development to clinical application is burdened by numerous challenges, including costs, regulatory approvals, data quality, reimbursement, and rigorous multi-step validations, which delay the adoption of DP into routine clinical practice. AI’s clinical adoption faces regulatory hurdles, including EU guidelines, CLIA certification, and FDA approval. DP also relies on high-quality data for AI training to provide a predictive performance [110], necessitating consensus in WSI references and multi-institutional validation. The validation of AI algorithms in DP is critical for ensuring accurate and reliable performance, which is essential for clinical decision-making. Validated algorithms not only comply with regulatory standards but also enhance patient trust and promote ethical practice. By assessing generalizability and robustness across diverse datasets and settings, validation studies ensure the applicability of algorithms in real-world clinical scenarios. Transparent reporting of validation results empowers clinicians to make informed decisions, driving ongoing quality improvement initiatives, which could ultimately enhance patient outcomes and healthcare delivery.

The successful incorporation of AI-DP into clinical practice also requires robust technical support to familiarize pathologists and clinicians. Also, addressing some pathologists’ hesitancy toward AI support can be mitigated through comprehensive training programs, emphasizing human-in-the-loop AI’s role in enhancing human expertise, rather than replacing it.

Difficulties in implementing the IS and other digital tools for colon cancer reporting include standardizing protocols for sample preparation and imaging, validating algorithms for accurate analysis, integrating DP with existing laboratory information systems, ensuring data interoperability, and addressing concerns regarding the reproducibility and clinical utility of digital biomarkers. Solutions involve interdisciplinary collaboration, stakeholder engagement, investment in infrastructure and training, regulatory compliance, and ongoing quality assurance measures.

Despite existing challenges, the introduction of AI-DP into the medical field shows significant potential for improving diagnostic accuracy and efficiency. Moreover, DP streamlines the daily workflow by offering numerous benefits. It facilitates telepathology, extending access to expert opinions on a global scale, with easy sharing of images, annotations, and diagnostic information, fostering interdisciplinary teamwork and improving the overall quality of patient care. Most importantly, DP allows to provide objective and quantitative biomarker data for clinical practice. Furthermore, AI-driven DP can improve pathology teaching for educational purpose. Therefore, it is imperative to systematically integrate AI algorithms, beginning with the validation of established biomarkers to ensure their alignment with existing diagnostic pathology practices. This will enhance the confidence in the clinical value of AI tools, paving the way for their seamless integration into clinical practice. As the implementation of AI advances, the inclusion of novel biomarkers will further enhance the role of DP as a crucial diagnostic tool in modern healthcare. The digital IS can be implemented by integrating image analysis software, specifically web-based software, that automatically quantifies immune cell infiltration in digital images, providing objective data to support clinical decision-making. In particular, the implementation of the IS in a digital workflow can vary depending on the system setup and preferences of the pathology department. It can be automatically generated alongside other diagnostic data or retrieved on demand by pathologists when needed for further analysis or decision support. The latter depends on factors such as the capabilities of the digital system, institutional protocols, and the preferences of pathologists and clinicians.

## 3. Discussion

DP outperforms traditional pathology practices by providing enhanced accuracy and facilitating remote collaboration among pathologists. In immuno-oncology, the integration of AI has been proven crucial for deciphering complex mechanisms within the TME, significantly impacting drug development.

In clinical settings, AI applications can also support oncologists in making personalized treatment plans, by improving diagnostic, prognostic, and predictive decision in cancer treatment.

IS is a DP approach that revolutionized clinical pathology by recognizing the immune response as a potent biomarker in cancer. IS demonstrated strong reproducibility, robustness, and standardized prognostic performance in assessing adaptive immune response in CRC. IS can also enhance diagnostic accuracy and classify patients based on immune parameters. Unlike visual assessment, IS-driven quantification of CD3+ and CD8+ cells on digital slide sections minimized the risk of patient misclassification [90], ensuring precise and consistent results in cancer diagnosis and prognosis.

Thus, IS demonstrated robustness while visual assessments lacked consistency [90]. As previously shown, the visual misclassification in pathology could lead to inaccurate prognosis and treatment decisions. This underscored the significant potential of the quantitative DP in clinical practice [89,90]. The visual assessment of TILs on H&E slides from CRC patients exhibited low concordance among pathologists [89,90]. Moreover, training pathologists did not significantly improve agreement, emphasizing the complexity and subjectivity of traditional pathology approaches. Additionally, misclassifying patients through visual pathology has profound clinical implications. For instance, misidentifying stage II CRC patients with high IS as low risk could lead to inadequate monitoring and treatment [90]. Conversely, misclassifying high IS stage II CRC patients as low risk might result in the administration of unnecessary chemotherapy [90].

Recently, the immune response criterion has gained large interest and was incorporated into the 5th edition of the WHO classification of digestive tumors, where it is regarded as an essential and desirable biomarker. Additionally, IS obtained a certificate from the CLIA program to perform testing on human specimens. IS has also found its way into the 2020 European Society for Medical Oncology (ESMO) guidelines and the 2021 Pan-Asian adapted ESMO Clinical Practice Guidelines [111,112], to enhance prognosis and consequently fine-tune the decision-making process for chemotherapy. While it has not been included in the National Comprehensive Cancer Network^®^ (NCCN^®^) guidelines yet, IS has proven to be a potent biomarker for predicting cancer recurrence, offering a novel approach in cancer treatment.

## 4. Conclusions

AI and computational biology advancements hold the potential to revolutionize cancer patient care, improving diagnosis, drug development, and precision medicine. Standardized IS has proven superior to visual assessments by expert pathologists in predicting risk of relapse [89,90]. This highlights the importance of implementing DP in clinical practice for personalized treatment of colon cancer patients and potentially other cancer types [89,90]. Given the clinical impact of IS and its prognostic significance in shaping cancer treatment strategies, its integration into routine practice will potentially contribute a significant stride in the cancer treatment field.

## 5. Patents

JG has patents associated with immune prognostic biomarkers and immunotherapies. CB is the co-inventor of a patent US20180322632A1 licensed to Ventana Medical Systems.

## Figures and Tables

**Figure 1 cancers-16-01686-f001:**
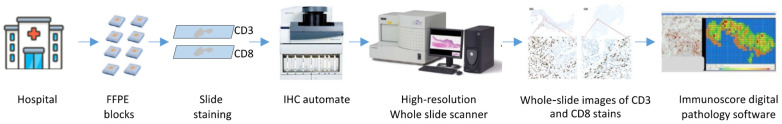
Immunoscore DP workflow for WSI quantitative analysis of immune response in cancer patients. Immunoscore^®^ analyzer software (INSERM/Veracyte, Marseille, France).

**Figure 2 cancers-16-01686-f002:**
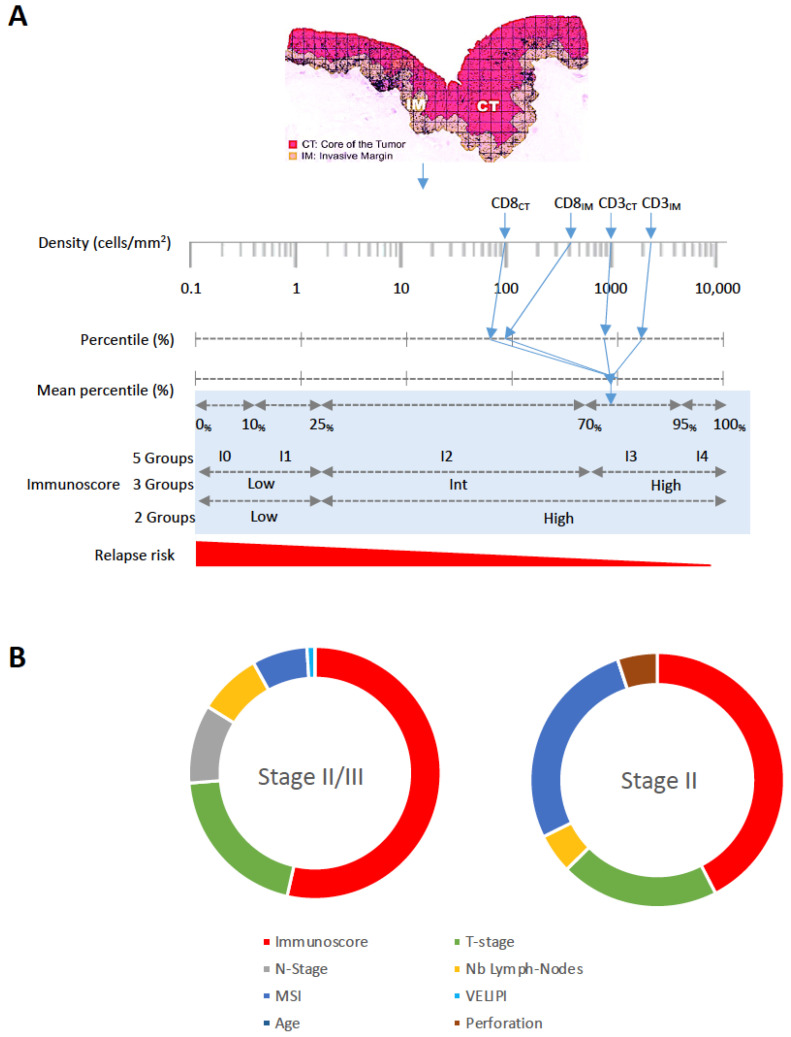
Illustration of the DP-Immunoscore calculation method. (**A**) Densities of CD3+ and CD8+ at both CT and IM are converted into percentile values. The mean percentile of the four markers is calculated and represented into a five-category (IS0, IS1, IS2, IS3, IS4) or a three- (IS-Low, IS-Int, IS-High) or a two-category scoring system (IS-Low, IS-High). Based on measuring immune response at the tumor site, IS predicts the risk of relapse in localized colon cancer to identify patients who could be spared from chemotherapy. (**B**) Ring charts illustrating the relative contribution of each risk parameter to recurrence risk in patients with stages II and II/III colon cancer. IS (red) is the highest predictor of time to recurrence (TTR) in both subgroups.

**Figure 3 cancers-16-01686-f003:**
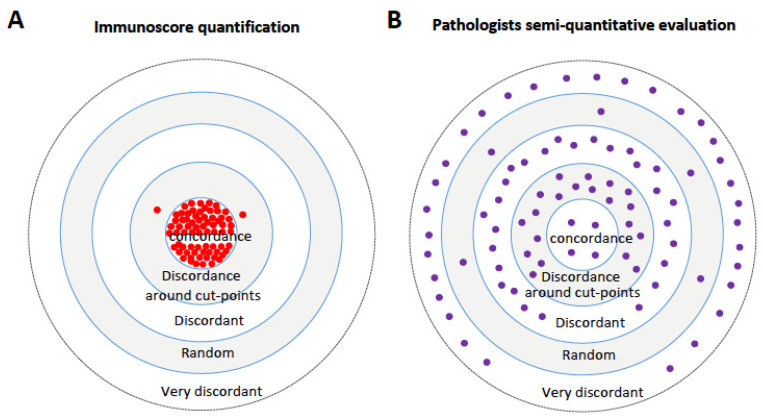
Target plot visualizations to depict the agreement between pathologists’ visual assessments and IS. These illustrations showcase the proportion of evaluations falling into different categories, including concordant, discordant around cut-points, discordant, very discordant (same case classified either High or Int or Low), and random cases (same case differently classified in 50% of evaluations). (**A**) IS repeated quantification, (**B**) the mean score of the pathologists’ evaluation of tumor-infiltrating lymphocytes (TIL) on hematoxylin–eosin (HE) slides and of pathologists’ evaluation of CD3 and CD8 stains before and after training. Each point in each plot represents assessments for four patients. Individual pathologist evaluation was previously published by Willis et al. Cancers 2023. Reprinted/adapted with permission from Ref. [90]. Copyright year, 2023 copyright owner’s J. Galon.

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
