# Peer review of "Digital Pathology for Better Clinical Practice"

_cancers, 2024, doi:10.3390/cancers16091686_

Round 1
Reviewer 1 Report
Comments and Suggestions for Authors
The paper is well written and well organised.
The topic is a very hot topic and could be usefull for the readers. It is incredibly clear from the review the importance of the Immunoscore, Immunotherapy evaluation of the tumour microenvironment (TME) The work is well written and well organised.
The topic is a very hot topic and could be useful to readers. From the review, the importance of the immunoscore, the assessment of the tumour microenvironment (TME) the immunotherapy and so on is very clear.
It is also clear the importance of the digital pathology.
What is missing, however, is "digital pathology for better clinical practise" in the day-to-day workflow of the pathology laboratory.
The authors should emphasise how digital pathology can be useful in the daily workflow. The authors should better explain how the "digital immunoscore" can be implemented. The authors should explain in a separate paragraph why DP implementation is still limited in a few hospitals. The authors should explain whether the immunoscore will be automatically presented to pathologists in a digital workflow or whether it will be retrieved on demand. The authors should identify difficulties and solutions on how to implement the Immunoscore and other in the everyday routine for colon cancer reporting Addressing all these points will be of added value for the paper.Author Response
Review: Digital Pathology for better clinical practice
Response to Reviewers:
Reviewer 1 comments
The paper is well written and well organised.
The topic is a very hot topic and could be usefull for the readers. It is incredibly clear from the review the importance of the Immunoscore, Immunotherapy evaluation of the tumour microenvironment (TME) The work is well written and well organised.
The topic is a very hot topic and could be useful to readers. From the review, the importance of the immunoscore, the assessment of the tumour microenvironment (TME) the immunotherapy and so on is very clear.
It is also clear the importance of the digital pathology.
What is missing, however, is "digital pathology for better clinical practise" in the day-to-day workflow of the pathology laboratory.
The authors should emphasise how digital pathology can be useful in the daily workflow. The authors should better explain how the "digital immunoscore" can be implemented. The authors should explain in a separate paragraph why DP implementation is still limited in a few hospitals. The authors should explain whether the immunoscore will be automatically presented to pathologists in a digital workflow or whether it will be retrieved on demand. The authors should identify difficulties and solutions on how to implement the Immunoscore and other in the everyday routine for colon cancer reporting. Addressing all these points will be of added value for the paper.
Response to Reviewer 1
We thank the reviewer for his interest in the topic and his nice review for the manuscript.
We have taken into consideration the relevant points addressed in the comments and responded accordingly.
1-The authors should emphasise how digital pathology can be useful in the daily workflow
Response: 357-368 lines ( in the manuscript)
DP streamlines the daily workflow by offering numerous benefits. It facilitates telepathology, extending access to expert opinions on a global scale, with easy sharing of images, annotations, and diagnostic information, fostering interdisciplinary teamwork and improving the overall quality of patient care. Most importantly, DP allows to provide objective and quantitative biomarker data for clinical practice. Furthermore, AI-driven DP can im-prove pathology teaching for educational purpose. Therefore, it is imperative to systematically integrate AI algorithms, beginning with the validation of established biomarkers to ensure their alignment with existing diagnostic pathology practices. This will enhance the confidence in the clinical value of AI tools, paving the way for its seamless integration into clinical practice. As the implementation of AI advances, the inclusion of novel biomarkers will further enhance the role of DP as a crucial diagnostic tool in modern healthcare.
2- The authors should better explain how the "digital immunoscore" can be implemented.
Response: lines 384-387 ( in the manuscript)
The digital IS can be implemented by integrating image analysis software, specifically web-based software, that automatically quantifies immune cell infiltration in digital images, providing objective data to support clinical decision-making.
3- The authors should explain in a separate paragraph why DP implementation is still limited in a few hospitals.
Response: lines 129-132 ( in the manuscript)
This can be attributed to factors such as high initial costs of infrastructure setup, concerns regarding data security and privacy, challenges in staff training and workflow integration, and regulatory hurdles related to validation and accreditation.
4- The authors should explain whether the immunoscore will be automatically presented to pathologists in a digital workflow or whether it will be retrieved on demand.
Response: lines 387-395 ( in the manuscript)
In particular, the implementation of the IS in a digital workflow can vary depending on the system setup and preferences of the pathology department. It can be automatically generated alongside other diagnostic data or retrieved on demand by pathologists when needed for further analysis or decision support. The latter depends on factors such as the capabilities of the digital system, institutional protocols, and the preferences of pathologists and clinicians.
5- The authors should identify difficulties and solutions on how to implement the Immunoscore and other in the everyday routine for colon cancer reporting .
Response: line 365-371 ( in the manuscript)
Difficulties in implementing the IS and other digital tools for colon cancer reporting include standardizing protocols for sample preparation and imaging, validating algorithms for accurate analysis, integrating DP with existing laboratory information systems, ensuring data interoperability, and addressing concerns regarding the reproducibility and clinical utility of digital biomarkers. Solutions involve interdisciplinary collaboration, stakeholder engagement, investment in infrastructure and training, regulatory compliance, and ongoing quality assurance measures.
Reviewer 2 Report
Comments and Suggestions for Authors
Hijazi et al., provided with a comprehensive review of the impact of Digital Pathology (DP) and Artificial Intelligence (AI) on cancer patient care, diagnosis and treatment. The authors effectively discussed the clinical importance of Standardized Immuno-score (SI) in predicting risk of relapse by expert pathologists, and highlighted the importance of DP application in clinical practice for personalized treatment of colon cancer and potentially other cancer types. The Review also emphasizes the potential of DP and AI in analyzing and interpreting high-volume data, aiding pathologists and oncologists in this process across the world.
The Review is well written and organized, and easy to read.
Minor comments:
1. It would be useful if the authors could provide with clinical examples of AI-personalized cancer diagnosis/treatment;
2. I would recommend the authors to briefly discuss the importance of the validation of algorithms for digital pathology;
3. since this is a Review, I would change the “Results” with another subtitle, as for example “Applications of AI and DP”;
4. keep consistency with the abbreviations across the text;
5. enlarge Figure 1.
Author Response
Reviewer 2 comments
Hijazi et al., provided with a comprehensive review of the impact of Digital Pathology (DP) and Artificial Intelligence (AI) on cancer patient care, diagnosis and treatment. The authors effectively discussed the clinical importance of Standardized Immuno-score (SI) in predicting risk of relapse by expert pathologists, and highlighted the importance of DP application in clinical practice for personalized treatment of colon cancer and potentially other cancer types. The Review also emphasizes the potential of DP and AI in analyzing and interpreting high-volume data, aiding pathologists and oncologists in this process across the world.
The Review is well written and organized, and easy to read.
Response to reviewer 2
We thank the reviewer for his thoughtful comments and his accurate description of the mansucript. We have taken into consideration his suggestions and responded/modified accordingly.
- Comment 1. It would be useful if the authors could provide with clinical examples of AI-personalized cancer diagnosis/treatment;
Response: lines 202-224 added to the manuscript.
AI-enabled personalized cancer diagnosis and treatment are reshaping oncology with impactful examples. The integration of AI into Glioblastoma imaging holds promise for advancing the characterization and tracking of the disease, including recurrence. This integration has the potential to substantially enhance patient outcomes through enabling more accurate diagnosis, precise treatment planning, and improved monitoring of treatment response63. In ovarian cancer, AI algorithms predict chemotherapy responses based on tumor genomic profiles, guiding clinicians in selecting the most effective treatment regimen64. Furthermore, in pancreatic cancer, AI is transforming pancreatic cancer care, enhancing diagnostics, personalizing treatments, and optimizing operational efficiency, ultimately leading to improved patient outcomes65. Additionally, AI-powered liquid biopsy platforms in CRC holds promise for screening, stratifying patients for treatment, and real-time monitoring of treatment response, offering improved management and personalized care66. These examples highlight how AI empowers clinicians with actionable insights, leading to more personalized and effective cancer care strategies.
- Comment 2. I would recommend the authors to briefly discuss the importance of the validation of algorithms for digital pathology;
Response: lines 351- 359 added to the manuscript.
The validation of AI algorithms in DP is critical for ensuring accurate and reliable performance, which is essential for clinical decision-making. Validated algorithms not only comply with regulatory standards but also enhance patient trust and promote ethical practice. By assessing generalizability and robustness across diverse datasets and settings, validation studies ensure the applicability of algorithms in real-world clinical scenarios. Transparent reporting of validation results empowers clinicians to make informed deci-sions, driving ongoing quality improvement initiatives, which could ultimately enhance patient outcomes and healthcare delivery.
- Comment 3. since this is a Review, I would change the “Results” with another subtitle, as for example “Applications of AI and DP”;
Response: The subtitle “Results” has been replaced accordingly, line 136.
- Comment 4. keep consistency with the abbreviations across the text;
Response: Abbreviations have been checked.
Comment 5. enlarge Figure 1.
Response: The figure has been enlarged.
Reviewer 3 Report
Comments and Suggestions for Authors
The manuscript reviews the future involvement of artificial intelligence in the diagnosis of cancer. After discovery of several biomarkers which modify the prognosis and treatment of the patients, pathology must perform more and more tests to diagnose the molecular subtypes of cancer tissue. Image analysis software has been used long time ago in microscope analyses, so the Digital Pathology is a natural step forward. The most important is the computer assisted quantitative analysis of whole – slide images since it enables to score much more data.
Pathologists try artificial intelligence, neural networks, and other software in image analysis in diagnosis of different cancer types demonstrating the potential of digital pathology in cancer diagnosis. The most important advantage of Digital Pathology is the possibility to analyse the whole section of tissue so it collects much more information which can be important for the diagnosis and treatment.
The authors show that digital pathology is better than traditional pathology and conclude that it should be implemented in routine diagnostics for the patients. Immunoscore biomarker becomes more and more important in clinical practice since immunotherapy as a targeted therapy is frequently used to treat patients with specific biomarkers.
Author Response
Reviewer 3 comments
The manuscript reviews the future involvement of artificial intelligence in the diagnosis of cancer. After discovery of several biomarkers which modify the prognosis and treatment of the patients, pathology must perform more and more tests to diagnose the molecular subtypes of cancer tissue. Image analysis software has been used long time ago in microscope analyses, so the Digital Pathology is a natural step forward. The most important is the computer assisted quantitative analysis of whole – slide images since it enables to score much more data.
Pathologists try artificial intelligence, neural networks, and other software in image analysis in diagnosis of different cancer types demonstrating the potential of digital pathology in cancer diagnosis. The most important advantage of Digital Pathology is the possibility to analyse the whole section of tissue so it collects much more information which can be important for the diagnosis and treatment.
The authors show that digital pathology is better than traditional pathology and conclude that it should be implemented in routine diagnostics for the patients. Immunoscore biomarker becomes more and more important in clinical practice since immunotherapy as a targeted therapy is frequently used to treat patients with specific biomarkers.
Response : We thank the reviewer for providing a precise description of the manuscript and for his time to review and evaluate the manuscript.